# Study on Dynamic Impact Response and Optimal Constitutive Model of Al-Mg-Si Aluminum Alloy

**DOI:** 10.3390/ma15217618

**Published:** 2022-10-30

**Authors:** Qinmin Zhang, Xiaomin Huang, Ran Guo, Dongyu Chen

**Affiliations:** Faculty of Civil Engineering and Mechanics, Kunming University of Science and Technology, Kunming 650500, China

**Keywords:** 6082-T6, dynamic impact response, adiabatic shear, SHPB, constitutive model

## Abstract

Al-Mg-Si series aluminum alloy is a heat-treatment-strengthened alloy. Research on the impact resistance of Al-Mg-Si series aluminum alloy is of great significance to expand its application in engineering. Taking 6082-T6 aluminum alloy as the concrete research object, using the split Hopkinson pressure bar (SHPB) device, the dynamic mechanical response of the material under different temperatures and average strain rates was studied, and the service performance of the material under extreme conditions was determined. The absolute temperature rise was introduced to optimize the existing constitutive model. The results show that when the environment temperature is 298.15~473.15 K under high-speed impact, the internal thermal softening effect of the material is dominant in the competition with the work hardening, resulting in a decrease in the flow stress of the material. Through the analysis of the real stress–strain curve, it was found that the elastic modulus of the material was negatively correlated with the strain rate, negatively correlated with the temperature, and showed an obvious temperature-softening effect. Yield strength was negatively correlated with temperature and positively correlated with strain rate, which showed an obvious strain rate hardening effect. Based on SEM microscopic analysis, it was found that under given conditions, adiabatic shear bands appeared in some samples, and their internal structures demonstrated obvious change. It was judged that when high-speed impact occurs, cracks are induced at the shear bands, and the cracks will continue to develop along the adiabatic shear bands, resulting in many oblique cracks which will gradually become larger and eventually lead to material failure. Finally, based on the model, the strain rate and temperature softening terms were improved, and a rise in adiabatic temperature rise was introduced. The improved model can better describe the strain rate effect of the material and accurately describe its flow stress. It provides a theoretical basis for the engineering application of materials.

## 1. Introduction

High strength, low density and corrosion resistance are some of the remarkable characteristics of aluminum alloy, and its good cold and hot working formability makes it the first choice in the field of lightweight materials in engineering [1]. The alloy known as 6082 aluminum alloy is an Al-Mg-Si alloy, which can be strengthened by heat treatment. Because of its excellent comprehensive properties, such as high specific strength, good machinability and strong corrosion resistance [2,3], it has been widely used in architecture, shipbuilding, high-speed trains, rail transit, automobiles, aerospace and other fields [4].

Experts and scholars at home and abroad have made important progress in researching the dynamic mechanical behavior of aluminum alloy [5,6,7,8,9,10,11,12,13,14,15,16,17]. For example, Deng Yunfei et al. [18] tested the mechanical properties of 6061-T651 aluminum alloy under different stress states, temperatures and strain rates, and found that when the material temperature is lower than 523.15 K, the temperature has a significant effect on the flow stress of the material, and the fracture strain is basically unaffected by the temperature. When the temperature of the material is higher than 523.15 K, the temperature has a significant influence on the flow stress and fracture strain of the material. The ballistic limit velocity and failure mode of the target plate were obtained by using a first-class air gun to test the impact of a spherical projectile on 6061-T651 aluminum alloy plate.

Li et al. [19] studied the effect of dynamic recrystallization (DRX) on the microstructure and mechanical properties of 6063 aluminum alloy extruded profiles, using experiments and simulations. The results show that the DRX score increases with the increase in temperature, strain rate and stamping speed. At high stamping speed, further increasing the stamping speed has much less effect on the temperature. Sun et al. [20] studied the effect of strain rate on quasi-static and dynamic mechanical properties and fracture behavior of 6005A-T6 aluminum alloy. The results show that the plasticity of 6005A-T6 aluminum alloy gradually increases with the increase in strain rate, but decreases when the strain rate reaches $200{\;S}^{-1}$. Under high-speed tensile deformation, the increase in dislocation density and slip band are the main reasons for the increase in strength and elongation at high speed.

Guo et al. [21] tested the dynamic mechanical properties of closed-cell aluminum foam under quasi-static and medium strain rate ($0.001\sim 100{\;S}^{-1}$). The results show that pure aluminum matrix foamed aluminum has no strain rate effect at medium and low strain rates, while foamed aluminum with high brittleness and low relative density has better energy absorption characteristics. The deformation bands of plastic and brittle matrix foamed aluminum show a “V” shape and an “X” shape, respectively, and brittle matrix foamed aluminum also has no strain rate effect. Khan et al. [22] found through studying the ballistic behavior of spray-formed Al-Zn-Mg-Cu alloy that the instantaneous temperature increase inside the material accompanied by local insulation, after the impact of a steel bullet, resulted in highly localized deformation (adiabatic shear band) and thermal softening, which macroscopically showed a decrease in material strength.

In addition, some scholars, such as Hu et al. [23], have studied the internal mechanism of the dynamic failure of aluminum alloy, and studied the microstructure evolution of 7A85 aluminum alloy under the conditions of strain rate ($0.0011{\;S}^{-1}$) and deformation temperature (523.15~723.15 K) via optical microscope (OM) and electron backscattering diffraction (EBSD). The results show that dynamic recovery (DRV) and dynamic recrystallization (DRX) are the main mechanisms of microstructure evolution during hot deformation of 7A85 aluminum alloy, and 623.15–673.15 K is the transition zone from dynamic recovery to dynamic recrystallization. At low temperature (623.15 K), DRV is the main mechanism, while DRX mainly occurs at high temperature (673.15 K). At this time, the sensitivity of microstructure evolution to temperature is relatively high. Yang et al. [24] studied the evolution mechanism of adiabatic shear band in 7075 aluminum alloy by combining thermodynamic analysis and numerical simulation, and pointed out that stress collapse first occurred at the position of soft second-phase particles and spread along the adjacent path of hard particles. Mondal et al. [25] confirmed that the fracture of 7075 aluminum alloy after ballistic impact was caused by the nucleation and development of microcracks in the adiabatic shear zone.

According to previous studies, different aluminum alloy nameplate numbers are accompanied by different impacts of mechanical behaviors. At present, the mechanical properties of aluminum alloy are mainly studied considering the coupling effect of ambient temperature and strain rate. In the analysis process, the temperature rise caused by the internal behavior of the material is not considered, and the constitutive models are all based on classical models. In the daily use of aluminum alloy, its mechanical properties will be affected by factors such as manufacturing process and service environment. Therefore, there is an urgent need to carry out mechanical properties analysis considering the coupling effect of absolute temperature rise and strain rate, and improve the classical constitutive model appropriately to obtain a constitutive model that is more in line with real-life applications of materials. In this paper, the medium-high-strength aluminum alloy 6082-T6 in Al-Mg-Si series aluminum alloy was used as the object, the split Hopkinson pressure bar (SHPB) was used to impact the sample at high speed and the microstructure was observed using a scanning electron microscope. The dynamic response behavior, microstructure evolution in impact process and dynamic impact optimization constitutive relation of materials are studied, which can provide a reference for the application of materials under extreme working conditions.

## 2. Methods

To test the dynamic mechanical properties of materials, the split Hopkinson pressure bar was used, and the structure diagram is shown in Figure 1. In the experiment, the specimen was placed between the incident bar and the transmission bar and the strain gauge was attached to the incident bar and the transmission bar in the way of 1/4 half-bridge circuit connection. The stress-wave signals in the test process were read according to the time sequence [26]. The strain rate $\overset{\cdot }{\varepsilon }\left(t\right)$, strain ε(t) and stress σ(t) of the material were obtained from the one-dimensional stress wave theory and the assumption of uniformity:(1)$$\overset{\cdot }{\varepsilon }\left(t\right)=-\frac{2C_{0}}{l_{s}}\varepsilon_{r}$$
(2)$$\varepsilon \left(t\right)=-\frac{2C_{0}}{l_{s}}\int_{0}^{t}\varepsilon_{r}\mathrm{dt}$$
(3)$$\sigma (t)=\frac{A}{A_{s}}E\varepsilon_{t}$$
where $l_{s}$ and $A_{s}$ as are the sample length and cross-sectional area, *A* and *E* are the cross-sectional area and elastic modulus of compression bar, $\varepsilon_{i}$ is the incident strain wave, $\varepsilon_{r}$ is the reflected strain wave, $\varepsilon_{t}$ is the transmission wave of the output rod and $C_{0}$ is the propagation speed of the stress wave.

According to the above formula, the stress–strain relationship can be obtained. In the dynamic compression experiment of 6082-T6 aluminum alloy, a cylindrical specimen with a dimension of φ Φ 5 mm × 5 mm was used. To ensure the accuracy of the experiment, the parallelism tolerance of both end faces of the specimen was kept at 0.001 [27]. The equipment was ZDSHPB-15. The length of incident rod, transmission rod, impact rod and absorption rod was 1200, 1400, 200 and 800 mm, the diameter was 16 mm, and the propagation rate of stress wave was 5100 m/s.

The experimental temperatures were set at 298.15, 323.15, 373.15 and 473.15 K, and the strain rate varied from $6\times 10^{-3}{\;S}^{-1}$ to $3\times 10^{3}{\;S}^{-1}$, in which the quasi-static compression was completed by an electronic universal testing machine. The elemental composition of Al-Mg-Si aluminum alloy is shown in Table 1. The experimental scheme is shown in Table 2. In order to avoid errors, the samples were tested several times in the same environment. The experimental test curves of the original signals of the incident wave and the transmitted wave collected by the strain gauge, and the incident signal $\varepsilon_{i}$, the reflected signal $\varepsilon_{r}$ and the transmitted signal $\varepsilon_{t}$ intercepted on the incident wave and the transmitted wave, are shown in Figure 2, all of which are displayed in the form of voltage.

## 3. Results

### 3.1. Flow Stress–Strain Relationship and Thermal–Mechanical Behavior under Impact Condition

The dynamic stress–strain relationship of the material was determined by processing and analyzing the experimental data (see Figure 3), in which Figure 3a is the real stress–strain curve with an ambient temperature of 473.15 K and strain rates of 1000, 2000 and $3000{\;S}^{-1}$, respectively; Figure 3b is the true stress–strain curves at the experimental temperatures of 298.15 K, 323.15 K, 373.15 K and 473.15 K, respectively and the strain rate of $3000{\;S}^{-1}$; Figure 3c is the true stress–strain curve when the experimental temperature is 323.15 K, 373.15 K and the strain rate is $1000{\;S}^{-1}$.

It can be seen from Figure 3a that at the same ambient temperature, the stress–strain curves of 6082-T6 aluminum alloy at different strain rates in the impact state show little change in the early elastic stage, but in the plastic stage, with the increase in strain rate, flow stress tends to rise and become larger. It can be seen from Figure 3b that under the same impact rate, with the increase in temperature, the flow–stress curves of 6082-T6 aluminum alloy all increase with the plastic strain, and the flow stress decreases. It shows that most of the plastic work carried out by the impact rod is converted into metal internal energy under high-speed impact conditions. With the influence of ambient temperature, the internal thermal softening effect of the material has the upper hand in the competition with work hardening (strain hardening). The 6082-T6 aluminum alloy is mainly composed of face-centered cubic crystals. During high-speed impact, the internal particles are dislocated, slipped, collided and extruded. With the increase in participating grains, dislocation and slip systems are induced, until all dislocation and slip systems in the sample participate together, and finally, a large amount of plastic deformation occurs.

Figure 4 shows changes in the elastic modulus and yield strength of samples under different conditions. It can be seen from Figure 4 that the elastic modulus of 6082-T6 aluminum alloy is negatively correlated with strain rate and temperature under dynamic impact load. Yield strength is negatively correlated with temperature and positively correlated with strain rate. The elastic modulus is a fixed property of materials, which changes with the change in atomic spacing. When the temperature rises, the atoms move actively and the spacing becomes larger, so the elastic modulus is negatively correlated with the temperature rise. With the increase in the strain rate, there is also a rise in strain value and yield stress, which indicates that 6082-T6 aluminum alloy has obvious strain rate effect under high-speed impact. The strain rate dependence of the material yield strength can be explained by the thermally activated dislocation theory. With the increase in temperature, the internal softening of the material leads to an increase in crystal spacing and dislocation deformation, resulting in a decrease in material flow stress, and thus, yield strength.

### 3.2. Dynamic Impact Constitutive Model

The accuracy of the analysis results is affected by the accuracy of the constitutive equation for simulating the dynamic response of materials. As far as the analysis of dynamic impact mechanical properties of materials is concerned, the Johnson–Cook model [28,29,30,31,32,33] is currently in widespread use. The model is constrained by five material constants which can reflect the influence of strain rate effect and temperature effect on the mechanical properties of materials. In this paper, the strain rate strengthening term and temperature softening term in the model are optimized to improve its accuracy. The expression of the Johnson–Cook dynamic response constitutive model [34] is as follows:(4)$$\sigma =(A+{B\varepsilon }_{P}{}^{n})(1+{\mathrm{Cln}\overset{\cdot }{\varepsilon }}^{*})\left(1-T^{*}{}^{m}\right)$$
where $\left(A+{B\varepsilon }_{P}{}^{n}\right),\;\left(1+{\mathrm{Cln}\overset{\cdot }{\varepsilon }}^{*}\right)$ and $\left(1-T^{*}{}^{m}\right)$ are descriptions of material hardening effect, strain rate effect and temperature softening effect. Select the reference strain rate and temperature (generally room temperature) where A is the yield strength, B is the hardening modulus, N is the hardening index, C is the strain rate sensitivity coefficient, and M is the material temperature softening index. $\varepsilon_{P}$ is equivalent plastic strain, $\overset{\cdot }{\varepsilon^{*}}=\overset{\cdot }{\varepsilon }/{\overset{\cdot }{\varepsilon }}_{0}$ is dimensionless plastic strain, ${\overset{\cdot }{\varepsilon }}_{0}$ is reference strain rate, $T^{*}=\frac{T-T_{r}}{T_{m}-T_{r}}$ is dimensionless temperature, $T_{m}$ is the melting point of the material (1023 K), and $T_{r}$ is room temperature (298.15 K).

According to the experimental data of quasi-static compression and dynamic impact, all parameters in the model are solved. The fitting of parameters A, B and N are all carried out under the conditions of reference strain rate and reference temperature. It can be determined from Formula (4) that when $T=T_{r}$ and $\overset{\cdot }{\varepsilon }={\overset{\cdot }{\varepsilon }}_{0}$, the constitutive model can be simplified into the following form:(5)$$\sigma =\left(A+{B\varepsilon }^{n}\right)$$

Generally, the temperature and strain rate of quasi-static compression experiment at room temperature are taken as reference values, and A is the yield strength of the material under this condition. Table 3 shows the yield strength of quasi-static experiment in this paper.

It can be seen from Table 3 that in the quasi-static test, when the strain rate is 0.6 × 10^−2^ S^−1^, the yield strength of the material is the lowest. In order to ensure the fitting accuracy, the temperature T = 298.15 K and the strain rate ${\overset{\cdot }{\varepsilon }}_{0}\;$= 0.6 × 10^−2^ S^−1^ are selected as the reference temperature and strain rate, and the parameters A and B can be fitted according to the σ−ε curve under the reference temperature and strain rate. Firstly, A can be determined, which is the yield strength of 233.4 MPa under this condition. Then, we take the logarithm of both sides of Formula (5) and bring A = 233.4 to obtain the relational expression:(6)ln(σ−233.4)=n×lnε+lnB

We make ln(σ−233.4)−lnε curve, then the parameter n represents the slope and lnB represents the intercept, and the values of b and n can be obtained. See Figure 5 for linear fitting.

When the test is at room temperature and the plastic strain is zero, the J-C constitutive model is:(7)$$\sigma_{y}=A\left(1+C\ln{\overset{\cdot }{\varepsilon }}^{*}\right)$$
where $\sigma_{y}$ is the yield stress of the material under quasi-static compression with low strain rate, which can be obtained from Table 3. We substitute the yield stress and strain rate of 6082-T6 aluminum alloy under static compression at room temperature into Formula (7) and draw an $\sigma_{y}-\ln{\overset{\cdot }{\varepsilon }}^{*}$ image, as shown in Figure 6. Parameter C can be fitted according to the method of fitting parameter N. According to the quasi-static and dynamic impact tests of 6082-T6, the related parameters of the constitutive model are shown in Table 4.

By substituting the values of all the parameters in Table 3 into Formula (4), the high-temperature dynamic impact J-C constitutive model of 6082-T6 aluminum alloy can be obtained as follows:(8)$$\sigma =(233.4+404\varepsilon_{P}{}^{0.74})(1+0.43{\ln\overset{\cdot }{\varepsilon }}^{*})(1-{\frac{T-T_{r}}{T_{m}-T_{r}}}^{1.83})$$

It can be seen from Equation (4) that the temperature-softening term in this model does not consider the absolute temperature rise in the material under high-speed impact. Under the impact of high strain rate, the deformation rate of the material is fast, internal energy is increased, and heat cannot rapidly be exchanged with the outside in a short time, resulting in an adiabatic environment. It has been found in [35,36] that when the material is subjected to high strain-rate impact, and the deformation condition is in an isothermal environment and has heat exchange with the outside world, internal temperature rise can be indirectly calculated by measuring the stored energy of the sample by calorimetry. At this time, the sample temperature is the sum of the experimental temperature and the absolute temperature rise.
(9)η∆W≈∆Q
(10)$$\eta \int_{0}^{\varepsilon }\sigma d\varepsilon ={\rho C}_{u}∆T$$
(11)$$∆T\left(\varepsilon \right)=\frac{\eta }{{\rho C}_{u}}\int_{0}^{\varepsilon }\sigma_{e}d\varepsilon$$

In the above formula, ∆W is the work achieved by compression deformation, ∆Q is the heat generated by compression, η is the industrial thermoplastic conversion coefficient, $\sigma_{e}$ is the true stress, ε is the true strain, $C_{u}$ is the specific heat at room temperature, ρ is the density, and ∆T is the absolute temperature rise.

It can be seen from Figure 5 that the relationship between $\sigma_{y}$ and $\ln{\overset{\cdot }{\varepsilon }}^{*}$ may not be completely linear. It is inappropriate to use a pure linear relationship to show their changing relationship. Rather, it is better to observe their changing trend and assume that they are exponential. The J-C constitutive model is modified by the above two points, as shown in Formula (12):(12)$$\sigma =(A+{B\varepsilon }_{P}{}^{n})\left\{1+e+\exp\left[\mathrm{Cln}\left(\frac{\overset{\cdot }{\varepsilon }}{{\overset{\cdot }{\varepsilon }}_{0}}\right)+d\right]\right\}(1-{\left(\frac{T+∆T-T_{r}}{T_{m}-T_{r}}\right)}^{m})$$

The experimental data are substituted into the optimized J-C constitutive model to fit the parameters. Figure 7 shows the relationship between improved $\sigma_{y}$ and $\ln{\overset{\cdot }{\varepsilon }}^{*}$ fitting, and the parameters of the improved model are shown in Table 5.

When the density of 6082-T6 aluminum alloy is 2.71 g/cm^3^, the specific heat at room temperature is 0.9 (J/kg·°C) and the work thermoplastic conversion coefficient of aluminum alloy is 0.9, the absolute temperature rise at various strain rates can be calculated by substituting the experimental data into Equation (11), as shown in Table 6.

Under the conditions of strain rates of 1000 S^−1^, 2000 S^−1^, 3000 S^−1^ and experimental environments of 323.15 K, 373.15 K and 473.15 K, the comparison between the fitting and experimental curves of the optimized J-C constitutive model considering absolute temperature rise is shown in Figure 8.

It can be seen from the comparison diagram in Figure 8 that the optimized model has a high degree of fitting with the real constitutive curve, which can reflect the stress–strain change trend of 6082-T6 aluminum alloy during dynamic compression by strain rate and absolute heating, and the competitive dynamic of strain hardening and high-temperature softening of the material at high strain rate. The optimized constitutive model is as follows:(13)$$\sigma =\left(233.4+404\varepsilon_{P}{}^{0.74}\right)\left\{0.97+\exp\;\left[0.81\times \ln\left(\frac{\overset{\cdot }{\varepsilon }}{{\overset{\cdot }{\varepsilon }}_{0}}\right)-3.1\right]\right\}(1-{\left(\frac{T+∆T-T_{r}}{T_{m}-T_{r}}\right)}^{1.83})$$

### 3.3. Micro-Morphology Analysis of Materials in Impact State

Adiabatic shear band generally occurs in the highly localized plastic deformation area [37] when the material is subjected to severe dynamic load or high temperature, and it is more common in metal materials. The deterioration of mechanical properties of materials caused by adiabatic shear band is an important form of malignant failure of materials.

All the samples observed by SEM are sampled along the axial direction of the SHPB test sample. The specific sampling method is shown in Figure 9. When the loading strain rate level is $6\times 10^{3}\sim 7.5\times 10^{3}{\;S}^{-1}$, the microstructure of the sample is shown in Figure 10.

As can be seen from Figure 10, after the SHPB experiment, the comparison of microstructures of three samples with room temperature and experimental conditions of 7000 S^−1^, 200 °C, 6000 S^−1^, 200 °C and 7000 S^−1^ showed few changes. However, the microstructure of the material changes greatly at room temperature and 6000 S^−1^, 100 °C and 6000 S^−1^, 100 °C and 7000 S^−1^, 200 °C and 7500 S^−1^. Shear bands appear in the four samples, as shown in the straight-line marked areas in Figure 10.

Adiabatic shear is related to the load and working temperature of the material, and high-speed impact and high temperatures lead to an increase in the strain rate of the material. For this experiment, the adiabatic shear phenomenon occurred at different temperatures and strain rates. The existing research shows that under high-speed impact loading, the deformed shear band (DSB) first appears in the material, and then with the increase in local strain, it further develops into the transformed shear band (TSB) [38]. Similar phenomena of the dissolution of the second- phase particles [39,40,41] have been found in the phase-transformation zones formed by dynamic impact loads of various high-strength aluminum alloys, and the microstructures of the deformation zones and the phase-transformation zones are obviously different. The crystalline structure is highly fragmented to form a deformation zone, and the center of the shear zone is moved slowly over the crystalline structure. There is almost no phase transition in the center of the zone, only the deformation streamline of the original structure. The phase-transition zone around the center of the belt is clearly demarcated from the deformation zone, and the bandwidth is complete.

In this experiment, the microstructure of the samples with adiabatic shear band was enlarged and analyzed. As shown in Figure 11, fine cracks appeared near the shear band (as shown in Figure 11c,d), and the internal part showed the morphological characteristics of shear deformation band (DSB) (the area between the two red curves in Figure 11a) and phase transformation band (TSB) (the area between the two red curves in Figure 11b. As shown in Figure 11b, the second-phase particles in the internal original matrix structure seemed to have partially dissolved in the phase-transformation band. The remaining undissolved second-phase particles extended along the path of the phase-change zone (direction of orange arrow line in Figure 11b). It can be judged that when high-speed impact occurs, cracks will be induced at the shear zone, and the cracks will develop continuously along the adiabatic shear zone, resulting in many oblique cracks.

## 4. Conclusions

Through analysis of the real stress–strain curve, it is founded that when the ambient temperature is 298.15~473.15 K and the strain rate is high, the dynamic mechanical response behavior of 6082 aluminum alloy shows that the internal thermal softening effect of the material is dominant in the competition with work hardening (strain hardening), resulting in a decrease in the flow stress of 6082 aluminum alloy, and the elastic modulus is negatively correlated with the strain rate and temperature. Yield strength is negatively correlated with temperature and positively correlated with strain rate.Based on the Johnson–Cook rheological stress–strain constitutive model, the strain- rate strengthening term and temperature softening term in the model are appropriately improved by introducing absolute heating, and the improved constitutive model is obtained as follows: $\sigma =(233.4+404\varepsilon_{P}{}^{0.74})\left\{0.97+\exp\;\left[0.81\times \ln\left(\frac{\overset{\cdot }{\varepsilon }}{{\overset{\cdot }{\varepsilon }}_{0}}\right)-3.1\right]\right\}(1-{\left(\frac{T+∆T-T_{r}}{T_{m}-T_{r}}\right)}^{1.83})$. By comparing the constitutive model with the experimental curve, it is found that the optimized model can fit well with the experimental curve, and it can also reflect the competitive dynamics of strain hardening and high temperature softening of materials at high strain rate. This verifies the accuracy of the model and provides a theoretical basis for the engineering application of materials.

Based on SEM microscopic analysis, it is found that under the conditions of this paper, some aluminum alloy samples have adiabatic shear. When the ambient temperature is 473.15 K and the average strain rate is $7500{\;S}^{-1}$, adiabatic temperature increases and the ambient temperature is superimposed, so that some second phases dissolve back at the local shear band, which leads to the fracture failure of the material when it is far below the phase-transition temperature. The appearance of shear band will induce cracks. It is speculated that adiabatic shear band is the main cause of material failure, and that the adiabatic shear phenomenon is linked to the coupling effect of temperature and strain rate.

## Figures and Tables

**Figure 1 materials-15-07618-f001:**
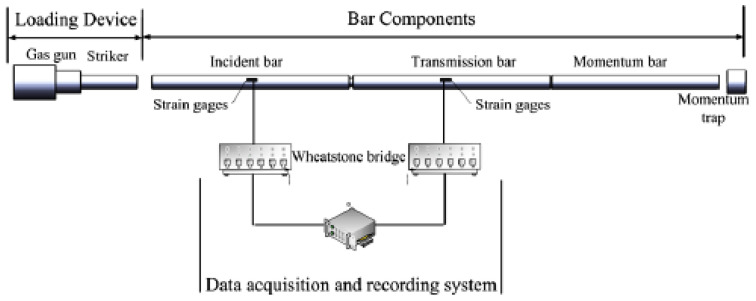
Schematic diagram of the device.

**Figure 2 materials-15-07618-f002:**
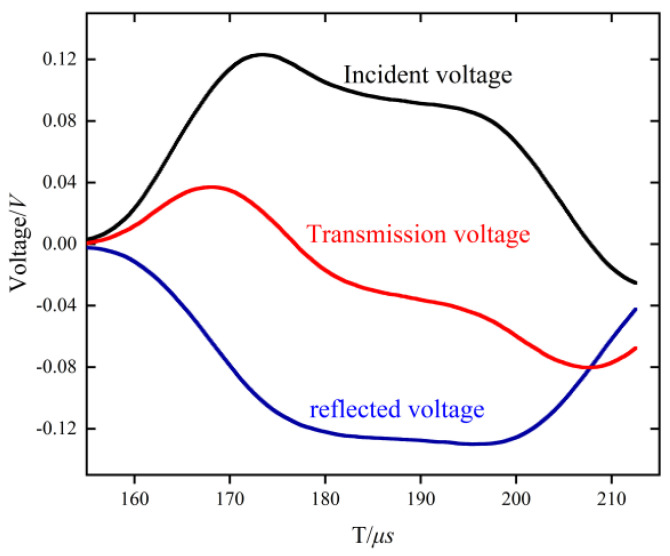
Original signals of incident wave, reflected wave and transmitted wave.

**Figure 3 materials-15-07618-f003:**
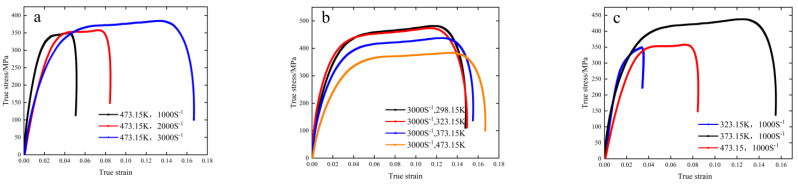
Real stress–strain curves under different test conditions. (**a**) 473.15 K. (**b**) 3000 S^−^^1^. (**c**) 1000 S^−^^1^.

**Figure 4 materials-15-07618-f004:**
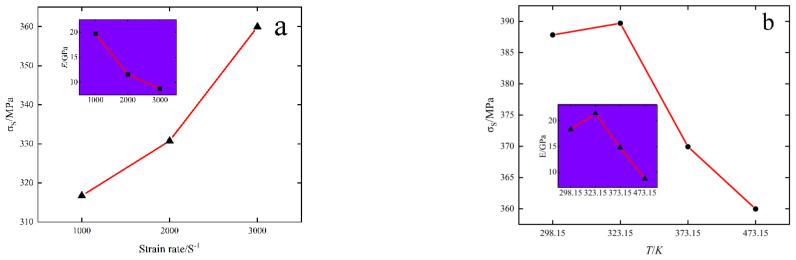
Changes in elastic modulus and yield strength under different conditions: (**a**) 473.15 K, (**b**) 3000 S^−1^.

**Figure 5 materials-15-07618-f005:**
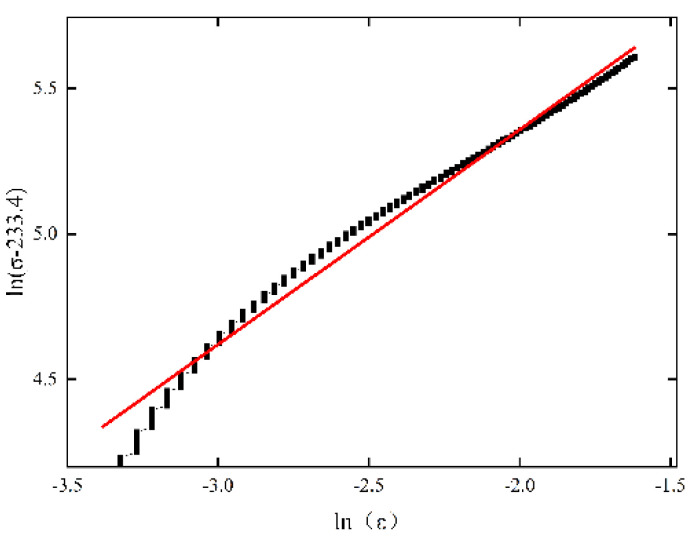
Linear fitting of parameters A, B and N.

**Figure 6 materials-15-07618-f006:**
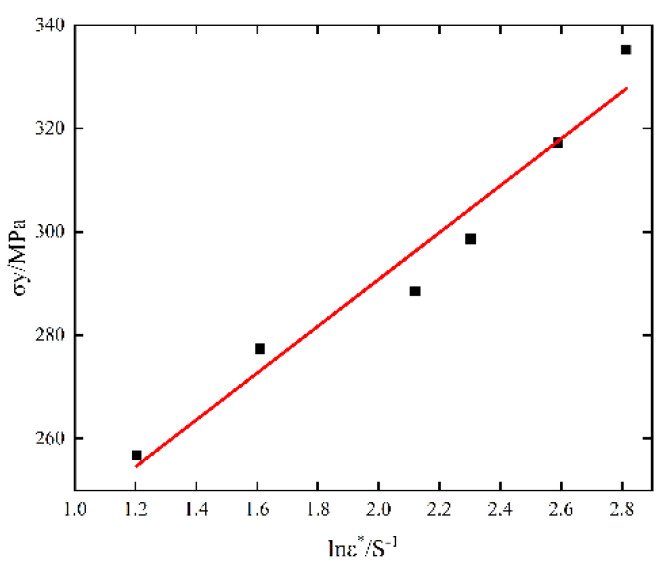
$\sigma_{y}-\ln{\overset{\cdot }{\varepsilon }}^{*}$ Fitting relationship.

**Figure 7 materials-15-07618-f007:**
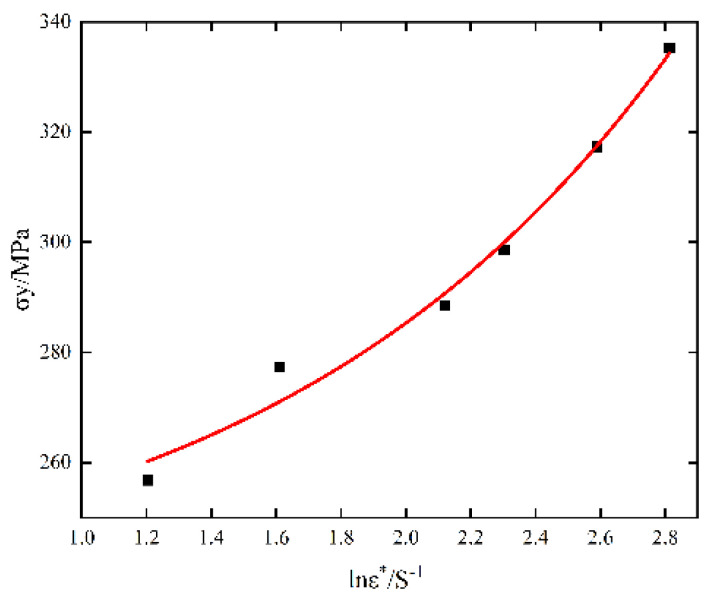
Improved $\sigma_{y}-\ln{\overset{\cdot }{\varepsilon }}^{*}$ fitting relationship.

**Figure 8 materials-15-07618-f008:**
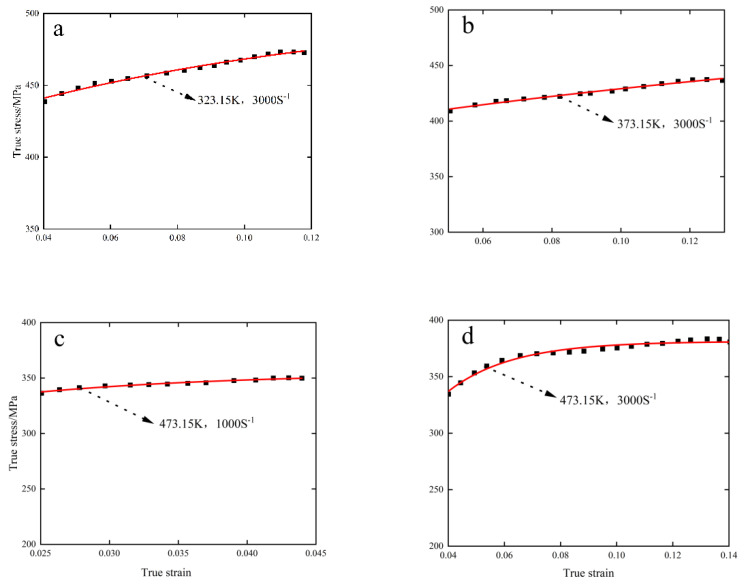
Comparison between experimental curve and fitting curve under different conditions. (**a**) 323.15 K, 3000 S^−^^1^. (**b**) 373.15 K, 3000 S^−^^1^. (**c**) 473.15 K, 1000 S^−^^1^. (**d**) 473.15 K, 3000 S^−^^1^.

**Figure 9 materials-15-07618-f009:**
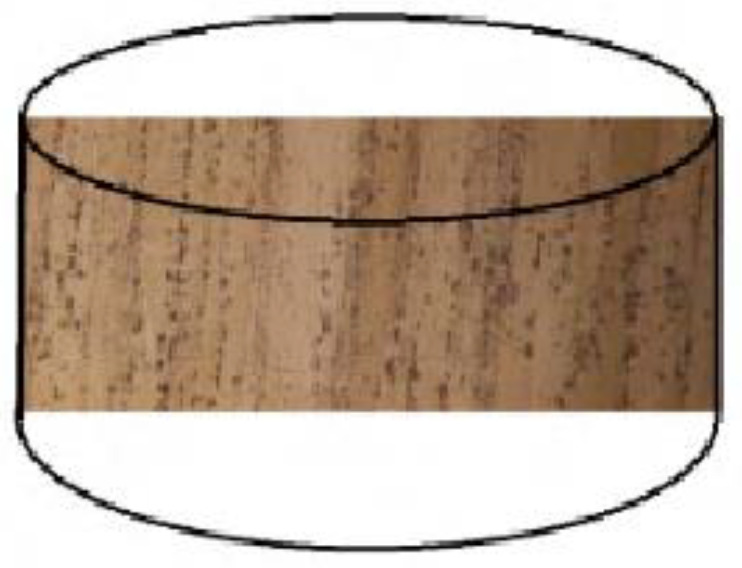
SEM sampling method.

**Figure 10 materials-15-07618-f010:**
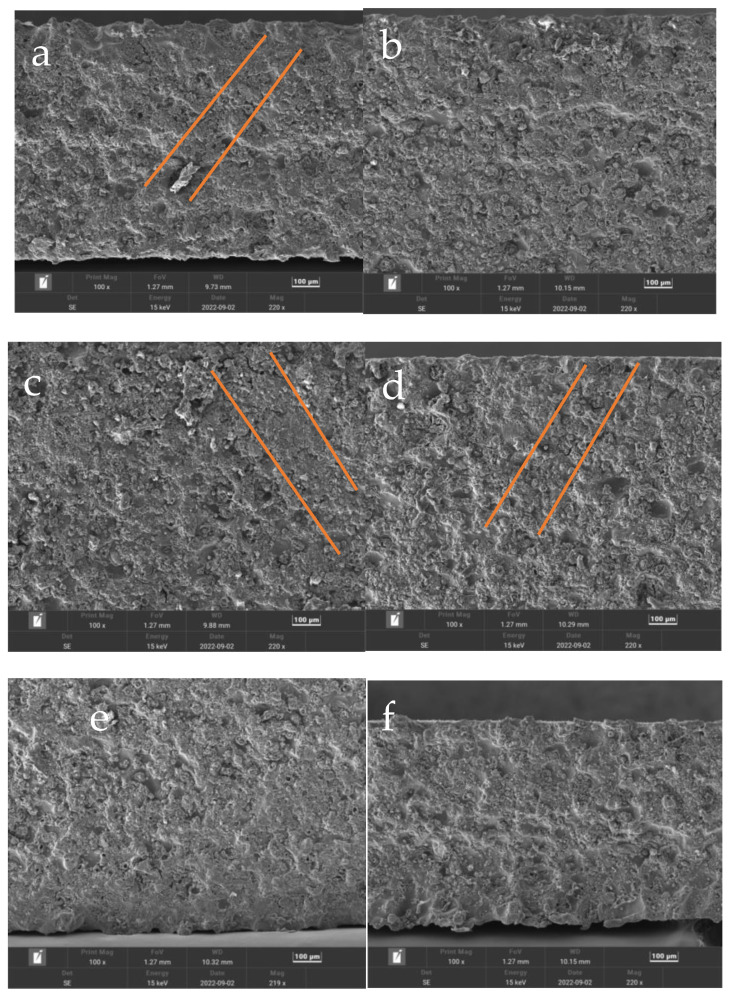
Microstructure of samples under different conditions: (**a**) 298.15 K, 6000 S^−1^, (**b**) 298.15 K, 7000 S^−1^, (**c**) 373.15 K, 6000 S^−1^, (**d**) 373.15 K, 7000 S^−1^, (**e**) 473.15 K, 6000 S^−1^, (**f**) 473.15 K, 7000 S^−1^, (**g**) 473.15 K, 7500 S^−1^.

**Figure 11 materials-15-07618-f011:**
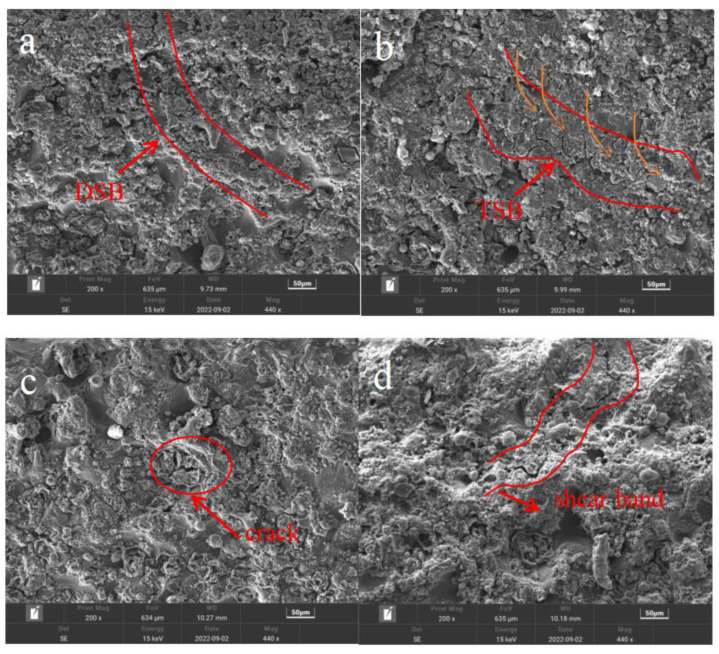
Partial enlargement of adiabatic shear band (**a**): 298.15 K, 6000 S^−1^, (**b**): 373.15 K, 6000 S^−1^, (**c**): 373.15 K, 7000 S^−1^, (**d**): 473.15 K, 7500 S^−1^.

**Table 1 materials-15-07618-t001:** Element Composition of Al-Mg-Si Aluminum Alloy Materials (%).

Si.	Fe	Cu	Mn	Mg	Cr	Zn	Ti	Al
0.7~1.3	0.50	0.10	0.4~1.0	0.6~1.2	0.25	0.20	0.10	Bal

**Table 2 materials-15-07618-t002:** Experimental scheme.

Dynamic Impact							Quasi-Static Compression	
Temperature/K	strain rate ($S^{-1}$)						1 × 10^−1^ S^−1^	Sample 1
	1000	2000	3000	6000	7000	7500	8 × 10^−2^ S^−1^	Sample 2
298.15			Sample 8	Sample 16	Sample 17		6 × 10^−2^ S^−1^	Sample 3
323.15	Sample 9		Sample 10				5 × 10^−2^ S^−1^	Sample 4
373.15	Sample 11		Sample 12	Sample 18	Sample 19		3 × 10^−2^ S^−1^	Sample 5
473.15	Sample 13	Sample 14	Sample 15	Sample 20	Sample 21	Sample 22	2 × 10^−2^ S^−1^	Sample 6
							6 × 10^−3^ S^−1^	Sample 7

**Table 3 materials-15-07618-t003:** Relevant parameters of quasi-static test.

	Strain Rate (10^−2^ S^−1^)						
	0.6	2	3	5	6	8	10
Yield strength (MPa)	233.4	256.8	277.4	288.5	298.6	317.3	335.3

**Table 4 materials-15-07618-t004:** J-C constitutive model parameters.

A/MPa.	B/MPa	C	n	m
233.4	404	0.43	0.74	1.83

**Table 5 materials-15-07618-t005:** Parameters of J-C model after optimization.

A/MPa.	B/MPa	C	n	m	e	d
233.4	404	0.81	0.74	1.83	−0.03	−3.1

**Table 6 materials-15-07618-t006:** Absolute temperature and theoretical temperature corresponding to each loading strain rate.

Strain rate (S^−1^)	1000	2000	3000		
Absolute warming (ΔT)/K	306.15	327.75	385.35	403.15	429.75
Ambient temperature (T)/K	473.15	473.15	323.15	373.15	473.25
Theoretical temperature (ΔT+T)/K	779.3	800.9	708.5	776.3	909.2

## Data Availability

This study does not use other people’s research data, and the data in this paper are all obtained through experiments.

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
