# Peer review of "Study on Dynamic Impact Response and Optimal Constitutive Model of Al-Mg-Si Aluminum Alloy"

_materials, 2022, doi:10.3390/ma15217618_

Round 1

Reviewer 1 Report

In this study, absolute temperature rise was introduced to optimize the existing constitutive model.

The results show that when the environment temperature is 298.15~473.15K under high-speed impact, the internal thermal softening effect of the material is dominant in the competition with the work hardening.

These results are very interesting, which is enough to be published in this journal.

Author Response

Response to Reviewer 1 Comments

First of all, thank you, teacher, for taking time out of your busy work to review my paper. As I didn't see your revision opinions in the journal official website, I revised my paper based on the opinions of the other three reviewers, and look forward to your review again. Finally, thank you again for your valuable suggestions. I wish you good health, smooth work and harmonious family.

Reviewer 2 Report

In the review of research article titled: Study on Dynamic Impact Response and Optimal Constitutive Model of Al-Mg-Si Aluminum Alloy, the authors have described the research work very well covering a lot of aspects. I would like to see this article publish but after some minor modifications as follow;

1.      Authors can make the abstract portion more attractive by highlighting their achieved results (values) of Real stress-strain curves, elastic modulus and yield strength.

2.      I would like to suggest the authors to discuss the drawbacks in the previous literature which has compelled the authors to carry out this study and specifically on the Al-Mg-Si Aluminum Alloys.

3.      Study will be more interesting if authors can add the elemental colored SEM images of the fabricated materials.

4.      In the end it will be better to compare the achieved Real stress-strain curves, elastic modulus and yield strength results in the form of table with the previously reported data.

5.      In conclusion please clearly describe the best merits of your study; too much literature in conclusion is making it un-clear. Please make it precise.

Best of luck.

Author Response

Response to Reviewer 2 Comments

1:Authors can make the abstract portion more attractive by highlighting their achieved results (values) of Real stress-strain curves, elastic modulus and yield strength.

Response 1:The abstract has been revised.

2:I would like to suggest the authors to discuss the drawbacks in the previous literature which has compelled the authors to carry out this study and specifically on the Al-Mg-Si Aluminum Alloys.

Response 2:I have summarized the literature discussed above, and put forward the research focus of this paper and the shortcomings of previous studies in 88 lines.

3:Study will be more interesting if authors can add the elemental colored SEM images of the fabricated materials.

Response 3:I have also considered using color SEM images before. Because the microscopic analysis of this study mainly discusses the accompanying phenomenon of adiabatic shear band and crack, if color images are used, the observation of crack generation is not obvious.

4:In the end it will be better to compare the achieved Real stress-strain curves, elastic modulus and yield strength results in the form of table with the previously reported data.

Response 4:For elastic modulus and yield strength, this paper mainly discusses the influence of temperature and strain rate on them, and Figure 4 analyzes the influence of temperature and strain rate on them. You put forward the revised opinion of comparing with the data of previous studies, and I consulted the relevant literature. Because of the different experimental conditions and materials, it is not easy to make a comparison.

5:In conclusion please clearly describe the best merits of your study; too much literature in conclusion is making it un-clear. Please make it precise.

Response 5:The conclusion has been revised.

Finally, thank you again for your valuable revision opinions and look forward to your review again. Here, I wish you good health, smooth work and family harmony.

Reviewer 3 Report

The paper reports on mechanical tests of Al-Mg-Si alloys, using a split Hopkinson pressure bar instrument. The experiments were performed carefully, and the results are meaningful: the authors investigated thermal softening and strain hardening of the material, resulting in the decrease of flow stress; further they determined the correlations between elastic modulus and yield strength as a function of temperature and strain rate. A new semiempirical equation was determined for stress-strain relationship. Adiabatic shear was observed at certain experimental conditions.

The paper is generally well and understandably written in good English. I only have some minor comments:

Line 50,51: … when the strain rate reaches. Everybody would ask: reaches what?

Line 114 and further: It looks the seconds are written as capital S. I would propose using small s.

Line 249: Figure 7 shows. – Figure 7 seems to be missing in the manuscript.

Line 284: did not change much, but room temperature and … – This sentence seems a bit invalid. I would expand as ... did not change much, but they change considerably at room temperature and …

Line 291, Figure 10 caption: Comment what the red lines show.

Line 302: is slowly over the matrix – is slowly moving over the matrix (?)

Author Response

Response to Reviewer 3 Comments

1:Line 50,51: … when the strain rate reaches. Everybody would ask: reaches what?

Response 1:The strain rate missing from the original 50 lines has been rewritten, as shown in line 58.

2:Line 114 and further: It looks the seconds are written as capital S. I would propose using small s.

Response 2:The capital S of the original 114 lines has been changed to lowercase S, see line 127.

3:Line 249: Figure 7 shows. – Figure 7 seems to be missing in the manuscript.

Response 3:Figure 7 has been added, see line 276.

4:Line 284: did not change much, but room temperature and … – This sentence seems a bit invalid. I would expand as ... did not change much, but they change considerably at room temperature and …

Response 4:The original 284 lines have been modified, see line 309.

5:Line 291, Figure 10 caption: Comment what the red lines show.

Response 5:The micro-analysis diagrams of Figure 10 and Figure 11 have been marked accordingly.

6:Line 302: is slowly over the matrix – is slowly moving over the matrix (?)

Response 6:The original 302 lines have been modified, see 330 lines.

Finally, thank you again for your valuable revision opinions and look forward to your review again. Here, I wish you good health, smooth work and family harmony.

Reviewer 4 Report

I have read the article titled " Study on Dynamic Impact Response and Optimal Constitutive Model of Al-Mg-Si Aluminum Alloy" for possible publication in the Journal Materials. My initial comments are highlighted in the manuscript. Kindly look into it. Currently, I recommend MAJOR revision for this submission

Author Response

Response to Reviewer 4 Comments

First of all, thank you, teacher, for taking time out of your busy work to give your valuable advice on my thesis. Based on your revision suggestions, I made the following revisions:

  1. For the deficiency of the introduction, I summarized the previous research and highlighted the research focus, as shown in line 88.
  2. Based on the experimental content of the fund project, selecting the temperature range in this paper is more in line with the project research.

3: The alloy composition is shown in Table 1, Row 142.

4: Picture 2 has been enlarged.

  1. In Figure 10, the area between two horizontal lines is shear band, see line 313.
  2. According to your suggestion, it is explained in detail in Figure 11, such as shear zone, deformation zone, phase change zone and crack, and the shear zone includes deformation zone and phase change zone. For your suggestion of marking the second phase particles, it is difficult to mark the particles in the actual diagram. I marked the second particle elongation direction in Figure 11(b), which is along the direction of the phase change zone.

Finally, thank you again for your valuable revision opinions and look forward to your review again. Here, I wish you good health, smooth work and family harmony.

Round 2

Reviewer 4 Report

I recommend acceptance of this manuscript